# Quantifying antibiotic impact on within-patient dynamics of extended-spectrum beta-lactamase resistance

**Rene Niehus[1][†]\*, Esther van Kleef[2], Yin Mo[1], Agata Turlej-Rogacka[3], Christine Lammens[3], Yehuda Carmeli[4], Herman Goossens[3], Evelina Tacconelli[5,6], Biljana Carevic[7], Liliana Preotescu[8], Surbhi Malhotra-Kumar[3], Ben S Cooper[1]**

[1]University of Oxford, Oxford, United Kingdom; [2]National Institute for Public Health and theEnvironment, Bilthoven, Netherlands; [3]University of Antwerp, Antwerp, Belgium; [4]Tel-Aviv University, Tel-Aviv, Israel; [5]University of Tuebingen, Tuebingen, Germany; [6]Infectious Diseases, University of Verona, Verona, Italy; [7]Clinical Centre of Serbia, Belgrade, Serbia; [8]Matei Balş National Institute for Infectious Diseases, Bucharest, Romania

**\*For correspondence:**
rniehus@hsph.harvard.edu

**Present address:** [†]Harvard T.H. Chan School of Public Health, Harvard University, Cambridge, United States

**Abstract** Antibiotic-induced perturbation of the human gut flora is expected to play an important role in mediating the relationship between antibiotic use and the population prevalence of antibiotic resistance in bacteria, but little is known about how antibiotics affect within-host resistance dynamics. Here we develop a data-driven model of the within-host dynamics of extended-spectrum beta-lactamase (ESBL) producing *Enterobacteriaceae*. We use bla$_{CTX-M}$ (the most widespread ESBL gene family) and 16S rRNA (a proxy for bacterial load) abundance data from 833 rectal swabs from 133 ESBL-positive patients followed up in a prospective cohort study in three European hospitals. We find that cefuroxime and ceftriaxone are associated with increased bla$_{CTX-M}$ abundance during treatment (21% and 10% daily increase, respectively), while treatment with meropenem, piperacillin-tazobactam, and oral ciprofloxacin is associated with decreased bla$_{CTX-M}$ (8% daily decrease for all). The model predicts that typical antibiotic exposures can have substantial long-term effects on bla$_{CTX-M}$ carriage duration.

## Introduction

Antibiotic use can increase resistance prevalence in a host population through multiple pathways (*Lipsitch and Samore, 2002*). It may: (i) affect the duration of resistance carriage and hence transmission potential; (ii) increase bacterial load of resistant organisms and thus increase transmission; or (iii) selectively suppress host microbial flora where resistance is lacking, which may reduce the potential for transmission of sensitive organisms and also render hosts more susceptible to acquiring resistant bacteria. As well as being important for understanding population dynamics, levels of intestinal resistance are also likely to be important from an individual patient perspective. It has been shown, for instance, that the digestive tract is the primary source of enterobacteria causing bloodstream infections in haematological patients, and a high abundance of beta-lactam resistant enterobacteria in the gut flora is predictive of a high risk of a corresponding drug-resistant bloodstream infection (*Woerther et al., 2015*). Moreover, colonization with extended-spectrum beta-lactamase (ESBL)-producing *Enterobacteriaceae* amongst patients receiving cephalosporin-based prophylaxis prior to colorectal surgery is associated with a more than two-fold increase in risk of surgical site infection (*Dubinsky-Pertzov et al., 2019*). Therefore, quantifying within-host selection dynamics should lead to a better understanding of both individual patient-level and population-level risks and benefits of antibiotic use.

**eLife digest** Bacteria that are resistant to antibiotics are a growing global health crisis. One type of antibiotic resistance arises when certain bacteria that can produce enzymes called extended-spectrum beta-lactamases (or ESBLs for short) become more common in the gut. These enzymes stop important antibiotics, like penicillin, from working. However, exactly which antibiotics and treatment durations contribute to the emergence of this antibiotic resistance remain unknown.

Now, Niehus et al. find certain antibiotics that are associated with an increase in the number of gut bacteria carrying antibiotic resistance genes for ESBL enzymes. First, rectal swabs collected from 133 patients from three European hospitals were analysed to measure the total gut bacteria and the number of genes for ESBL enzymes. These samples had been collected at several time points including when the patient was first admitted to hospital, then every two to three days during their stay, and finally when they were discharged.

Combining the analysis of the samples with details of the patients' charts showed that treatment with two antibiotics: cefuroxime and ceftriaxone, was linked to an increase in ESBL genes in the gut bacteria. Other antibiotics – namely, meropenem, piperacillin-tazobactam and oral ciprofloxacin – were associated with a decrease in the number of bacteria with ESBL genes. Niehus et al. then performed further analysis to see if different treatment regimens affected how long patients were carrying gut bacteria with ESBL genes. This predicted that a longer course of meropenem, 14 days rather than 5 days, would shorten the length of time patients carried ESBL-resistant bacteria in their guts by 70%, although this effect will likely depend on the location of the hospital and the local prevalence of other types of antibiotic resistance.

This analysis reveals new details about how antibiotic treatment can affect ESBL resistance genes. More studies are needed to understand how antibiotics affect other antibiotic resistance genes and how resistant bacteria spread. This will help scientists understand how much specific antibiotic regimens contribute to antibiotic resistance. It may also help scientists develop new antibiotic treatment strategies that reduce antibiotic resistance.

Here we focus on *Enterobacteriaceae*, a bacterial family that is commonly found in the healthy mammalian gut microbiome (*Donnenberg, 1979*). Some member genus-species—*Klebsiella pneumoniae*, *Escherichia coli*, *Enterobacter spp.*—are important opportunistic human pathogens that can cause urinary tract, bloodstream, and intra-abdominal infections, as well as hospital-acquired respiratory tract infections. A major concern is the global increase in extended-spectrum beta-lactamase-producing organisms in this family (*Coque et al., 2008*; *Tacconelli et al., 2018*; *Valverde et al., 2004*). ESBL genes – of which the most important and globally widespread is the $bla_{CTX-M}$ gene family – confer resistance to clinically important broad-spectrum antimicrobials, such as third generation cephalosporins (*Paterson, 2000*). These genes commonly reside on large conjugative plasmids (*Bonnet, 2004*), and are co-carried with other antibiotic resistance determinants, making them a good marker for multi-drug resistance (MDR) in strains of *Enterobacteriaceae* (*Schwaber et al., 2005*). Because *Enterobacteriaceae* have their main biological niche in the gut microbiome (*Masci, 2005*), these bacteria are exposed to substantial collateral selection from antibiotics used to treat or prevent infections with other organisms ('bystander selection' [*Tedijanto et al., 2018*]). Quantifying the effects of antibiotic therapy on within-host resistance dynamics will help us to better understand the potential for selection of drug-resistant *Enterobacteriaceae* associated with different patterns of antibiotic usage.

In this work, we analysed sequential rectal swabs (n = 833) from 133 ESBL positive hospitalised patients from three hospitals (Italy, Romania, Serbia) to study the dynamics of antibiotic resistance gene abundance. Both $bla_{CTX-M}$ gene and, as a proxy for total bacterial load, 16S rRNA gene abundance were determined using quantitative polymerase chain reaction (qPCR). Previously, using a subset of these data, Meletiadis et al. demonstrated a statistical association between exposure to ceftriaxone and increases in $bla_{CTX-M}$ normalised by total bacterial load. Here, we addressed some broader questions. We studied the effects of a range of different antibiotics on the abundance of $bla_{CTX-M}$ and of 16S rRNA, and we aimed to fully characterise the within and between host variation of $bla_{CTX-M}$ and 16S rRNA and their within-host dynamics. For this purpose we developed a novel

dynamic model, a state-space model that we fit to fine-grained patient-level measurements and antibiotic exposure data. By incorporating hidden-state dynamics our model allowed us to dissect and quantify different types of data variability, such as noise from qPCR measurement or from the DNA extraction process, and to separate this from the within-host processes. In this way we directly estimated ecologically important parameters such as strength of resistance amplification during antibiotic treatment or the rate of decline of $bla_{CTX-M}$. We then used our model to make counterfactual predictions about how alternative choices of treatment would impact $bla_{CTX-M}$ carriage duration. The development of this data-driven within-host model and its use in exploring the impact of antibiotic treatment on amplification and loss of resistance is an important step in furthering our quantitative mechanistic understanding of how antibiotic use drives changes in the prevalence of resistance in a population.

## Results

### Patient cohort and treatment

The study enrolled a total of 1102 patients who were screened positive for ESBL producing *Enterobacteriaceae* at admission, and 133 patients (12%) gave consent to be included in the study: 51 (38%) from Romania; 52 (39%) from Serbia; and 30 (23%) from Italy. The median age was 59 years (range of 23–88), and 46% were female. The median length of hospital stay was 15 days (maximum of 53 days). All patients apart from one had two or more rectal swabs taken, with a median of five swabs per patient (range of 1–15). 114 out of 133 (86%) enrolled patients received antibiotics during their stay and 85 of these 114 (75%) received two or more different antibiotics, which were given both in mono- and combination therapy (see *Figure 1*). A total of 3993 patient days were observed, of which 2686 (67%) were days with antibiotic therapy (mono- or combination therapy). *Table 1* summarises important details of the study. Note that the antibiotics that we considered in this study were exclusively antibacterial drugs, and we ignored treatment with anti-tuberculosis drugs (pyrazinamide and isoniazid), which occurred only in two patients.

The different antibiotic classes, ranked by proportion of treatment days, were cephalosporins 25% ), fluoroquinolones (18%), penicillins (9%), nitroimidazole derivatives (metronidazole) (9%), glycopeptides (8%), carbapenems (5%), and others (26%). Two thirds of antibiotic treatment days were from intravenously administered antibiotics and one third from oral administration. Details on individual antibiotics are given in *Table 2*.

### Resistance dynamics

The time-varying $bla_{CTX-M}$ abundance exhibits a diverse range of dynamic patterns, including monotonic increases and decreases, as well as highly variable non-monotonic behaviour (*Figure 1*). Qualitatively similar fluctuations in $bla_{CTX-M}$ abundance were seen both in the presence and absence of antibiotic treatment. To determine whether this high level of dynamic variation contained a meaningful biological signal, we first studied temporal autocorrelation. If the observed variability is driven by observation uncertainty – for instance through the swab procedure, DNA extraction, or qPCR process – we expect autocorrelation close to zero in the time series. Conversely, if the observed fluctuations reflect true within-host dynamics in carriage levels, we would generally expect to see positive autocorrelation. We found a clear signal of first-order autocorrelation for both the $bla_{CTX-M}$ and the 16S rRNA gene time series, though autocorrelation was substantially stronger for the $bla_{CTX-M}$ data (*Figure 1—figure supplement 1a and b*). Using a Bayesian state-space model that decomposes the time series data into an observation component (representing noise due to variability in qPCR runs, and in the procedure of swab taking and sample processing) and a process component (due to the within-host dynamics), we estimated that much of the variability in $bla_{CTX-M}$ and 16S rRNA outcomes was due to measurement error associated with the swab procedure (median estimate of the proportion of total abundance variability attributable to swab error [90% credible interval [CrI]] of 54% [44%, 57%] and 73% [68%, 77%], respectively) (see *Figure 1—figure supplement 1c*).

However, the $bla_{CTX-M}$ data in particular were found to also contain a strong process component signal, indicating that a median estimate of 36% (90% CrI 30%, 43%) of the variability in the qPCR outcomes was due to underlying within-host dynamics (*Figure 1—figure supplement 1c*). To further investigate the determinants of $bla_{CTX-M}$ gene variation, we explored how much the $bla_{CTX-M}$ gene

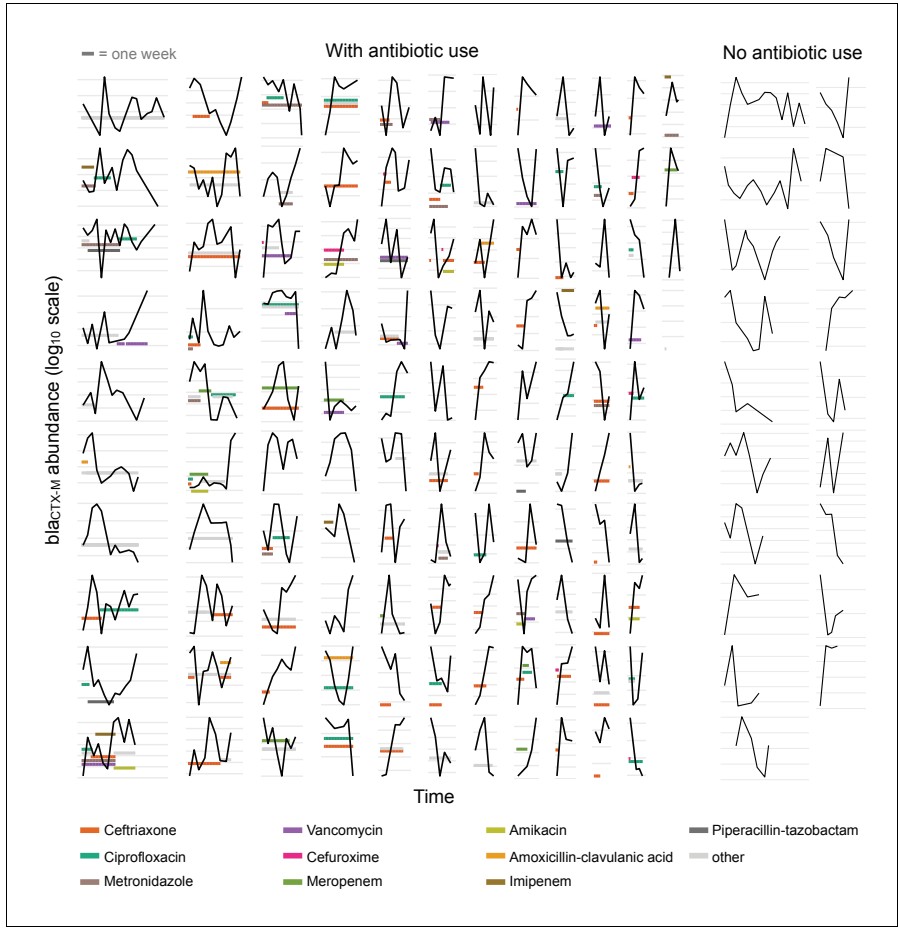

**Figure 1.** Time series plots demonstrating the diverse range of dynamical patterns of bla$_{\text{CTX-M}}$ resistance gene abundance across the 132 patients with two or more samples. The x-axis scale is identical across panels, the length of one week is given for scale in the top-left corner. Timelines are ordered by length. The y-axis scale differs between panels, with the space between vertical grey lines representing a 10-fold change in the absolute bla$_{\text{CTX-M}}$ gene abundance (measured in copy numbers). The left-hand side shows patients who received antibiotic treatment (n = 113), and the two right-hand side columns are patients without antibiotic treatment (n = 19). For clarity, we show only the twelve most frequently used antibiotics in distinct colours and other antibiotics in light grey.

The online version of this article includes the following figure supplement(s) for figure 1:

**Figure supplement 1.** Exploration of autocorrelation and different sources of data variability.

load varied between different patients or, over time, within the same patient. Using a Bayesian state-space model (see Methods and Materials) we found 16S rRNA gene abundance to be two orders of magnitude higher than bla$_{\text{CTX-M}}$ (median ratio 16S / bla$_{\text{CTX-M}}$ [90% CrI] 158 [88, 181]), with an estimated coefficient of variation (ratio of standard deviation to the mean) of 5.5 for 16S rRNA and 32.1 for bla$_{\text{CTX-M}}$. Between-patient abundance of bla$_{\text{CTX-M}}$ showed substantially more variability than within-patient abundance (median ratio [90% CrI] 134 [18, 1422]). In contrast, 16S rRNA gene abundance had similar between-patient and within-patient variability (median ratio [90% CrI] 0.8 [0.4,1.7]) (*Figure 2*). We noted that the rank plots (*Figure 1—figure supplement 1a*) indicate some convergence problems of $\sigma_{bio,16S}$, but several independent runs of the MCMC algorithm with different initial values consistently arrived at the same mean and standard deviation of the posterior estimate.

**Table 1.** Summary of the study.

| Number of participating hospitals | 3 (Serbia, Italy, Romania) |
|---|---|
| Study duration | 2 years (Jan 2011 - Dec 2012) |
| Inclusion criteria | Inpatients of medical and surgical wards, adults, non-pregnant, ESBL producting Enterobacteriaceae carriers (at admission) |
| Number of patients followed up | 133 (including one with a single swab taken) |
| Intervals between rectal swabs | two to three days |
| qPCR targets | $bla_{CTX-M}$ (ESBL resistance gene), 16S rRNA (total bacterial load) |
| Number of different antibiotics used | 35 |

This study (registration number NCT01208519) was conducted by the SATURN consortium, supported by the European Commission under the 7th Framework Program.

## Associating resistance and antibiotic treatment

The change in relative resistance between samples, measured as $bla_{CTX-M}$ abundance divided by 16S rRNA gene abundance, was only slightly elevated in time intervals where antibiotics were given compared to those where they were not (*Figure 3a*).

However, use of antibiotics with activity against *Enterobacteriaceae* to which carriage of $bla_{CTX-M}$ does not confer resistance (colistin, meropenem, ertapenem, imipenem, amoxicillin-clavulanic acid, ampicillin-sulbactam, piperacillin-tazobactam, gentamicin, amikacin, ciprofloxacin, ofloxacin, levofloxacin, tigecycline, doxycycline) was associated with a modest decrease in $bla_{CTX-M}$ abundance (*Figure 3b*). In contrast, the use of antibiotics with broad spectrum killing activity and to which carriage of $bla_{CTX-M}$ does confer resistance (cefepime, ceftazidime, ceftriaxone, cefotaxime, cefuroxime, amoxicillin, ampicillin) was associated with substantially higher increases in relative $bla_{CTX-M}$ abundance (*Figure 3c*).

## Dynamic antibiotic effect model

Fitting a dynamic model of $bla_{CTX-M}$ abundance and 16S rRNA abundance to the data (133 patients, 833 swabs, 3361 qPCR measurements), we found that cefuroxime and ceftriaxone were associated with increases in both absolute $bla_{CTX-M}$ abundance (mean daily increase [90% CrI] 21% [1%, 42%] and 10% [4%, 17%], respectively) and relative $bla_{CTX-M}$ abundance (14% [-1%, 30%] and 11% [5%, 17%], respectively) (*Figure 4*). Piperacillin-tazobactam, meropenem and ciprofloxacin (when given

**Table 2.** Overview of antibiotic treatments showing the ten most used antibiotics in this patient cohort.

Intravenous (iv), oral (or), and intramuscular (im) route administration is given in percent of treatment days.

| Antibiotic | Number of treated patients (total n=133) | Route |
|---|---|---|
| Ceftriaxone | 64 | ( 98% iv , 2% im ) |
| Ciprofloxacin | 34 | ( 67% iv , 33% or ) |
| Metronidazole | 20 | ( 50% iv , 50% or ) |
| Cefuroxime | 13 | ( 100% iv , 0% or ) |
| Vancomycin | 13 | ( 86% iv , 14% or ) |
| Meropenem | 10 | ( 100% iv , 0% or ) |
| Amikacin | 9 | ( 100% iv , 0% or ) |
| Amoxicillin-clavulanic acid | 9 | ( 57% iv , 43% or ) |
| Piperacillin-tazobactam | 7 | ( 100% iv , 0% or ) |
| Imipenem | 5 | ( 100% iv , 0% or ) |

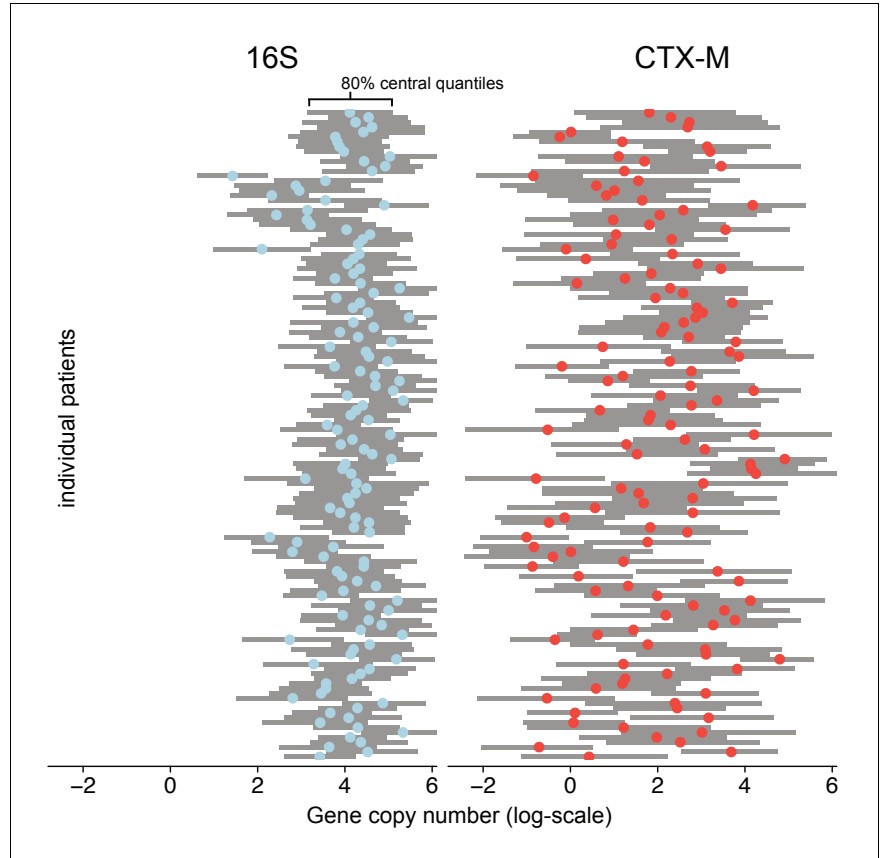

**Figure 2.** Variability of 16S abundance and bla$_{CTX-M}$ abundance within and between patient time series using a Bayesian hierarchical model. In this model, abundance is distributed around individual patient intercepts, which are distributed around a common population intercept. The plot shows individual patient intercepts given as mean posterior estimates (coloured dots) together with posterior predictions for sequence abundance for each patient (grey bars show 80% central quantiles).

The online version of this article includes the following figure supplement(s) for figure 2:

**Figure supplement 1.** Diagnostic plots of MCMC samples.

, 1%], and −8% −7%, 1%], and fect on relative iperacillin-tazo-). Intravenously d similar effects ole, and amoxi- oximated leave- or the dynamic ntibiotic effects

l for the bla$_{CTX-}$ del a threshold below which the bla$_{CTX-M}$ gene cannot be detected (see Materials and methods). Below this threshold the gene may either be lost from the bacterial community, or it may exist in very small reservoirs for example in persister cells (*Balaban et al., 2019*). The predictions of detectable carriage duration show a high degree of uncertainty, visible as long-tailed predictive distributions (*Figure 5*). Because of the skew, we report here the median instead of the mean together with 80% credible intervals. We chose the duration of different antimicrobial therapies according to clinical guidelines. Assuming

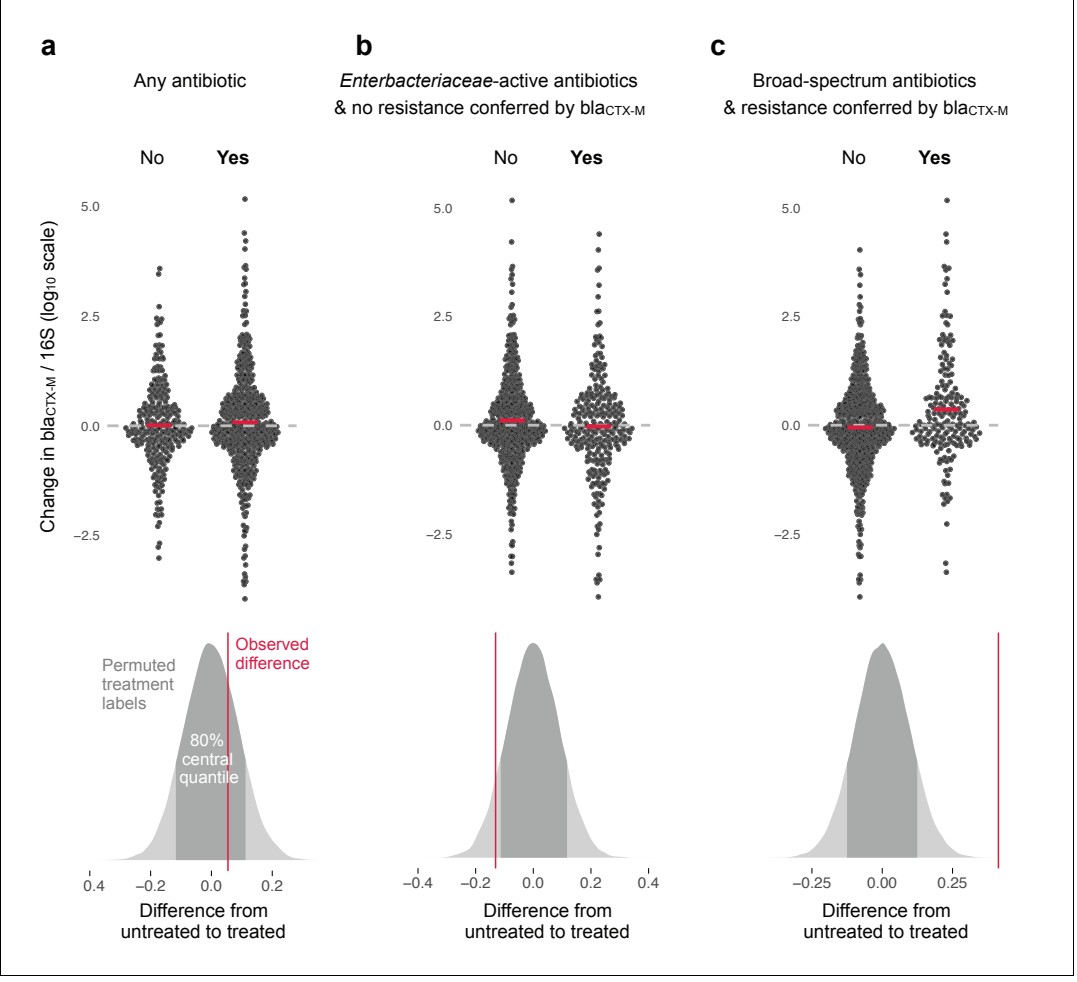

**Figure 3.** Association of antibiotic use with change in relative resistance (abundance of bla_CTX-M divided by abundance of 16S rRNA). The upper panels show the change in relative resistance between all neighbouring timepoints (black dots), dashed horizontal lines in grey indicate the region of no change. Pairs of violin scatter plots (with the mean values shown as red bars) contrast different treatment that occurred between those timepoints. 'Yes' indicates treatment with specified antibiotics and 'No' means treatment with other antibiotics or no treatment. The lower three panels show the distribution of mean differences of the change in relative resistance between treatment groups generated through treatment-label permutation (areas in darker grey show 80% central quantiles). The distributions are overlaid with the observed difference (red vertical line). Panel (a) compares treatment with any antibiotic versus no antibiotic (number of intervals without treatment/number of intervals with treatment are 251(N)/449(Y)). Panel (b) compares treatment with antibiotics with activity against *Enterobacteriaceae* and to which bla_CTX-M does not confer resistance (colistin, meropenem, ertapenem, imipenem, amoxicillin-clavulanic acid, ampicillin-sulbactam, piperacillin-tazobactam, gentamicin, amikacin, ciprofloxacin, ofloxacin, levofloxacin, tigecycline, doxycycline) with all other treatment, including no treatment (445(N)/255(Y)). Finally, in panel (c) we consider antibiotics with broad-spectrum activity but to which bla_CTX-M does confer resistance (cefepime, ceftazidime, ceftriaxone, cefotaxime, cefuroxime, amoxicillin, ampicillin) (513(N)/187(Y)).

that the estimated antibiotic associations represent causal effects, we find that a single eight day course of cefuroxime or a 14 day course of ceftriaxone substantially prolongs carriage of bla_CTX-M, by a median estimate of 147% (80% CrI 13.4%, 577%) for cefuroxime and 120% (80% CrI −8.6%, 492%) for ceftriaxone versus no exposure (*Figure 5*). Addition of oral ciprofloxacin to a course of amoxicillin-clavulanic acid or ceftriaxone reduces bla_CTX-M carriage duration (by approximately 51% [80% CrI −115%, 89%] and 48% [80% CrI −71.1%, 86%]) (*Figure 5*). A typical 14 day course of meropenem or a 8 day course of piperacillin-tazobactam reduce bla_CTX-M carriage duration relative to no treatment (by approximately 42% [80% CrI −25%, 75%] and 41% [80% CrI −45%, 71%], respectively), and each reduces bla_CTX-M carriage even more relative to a 7 day course of combined ceftriaxone

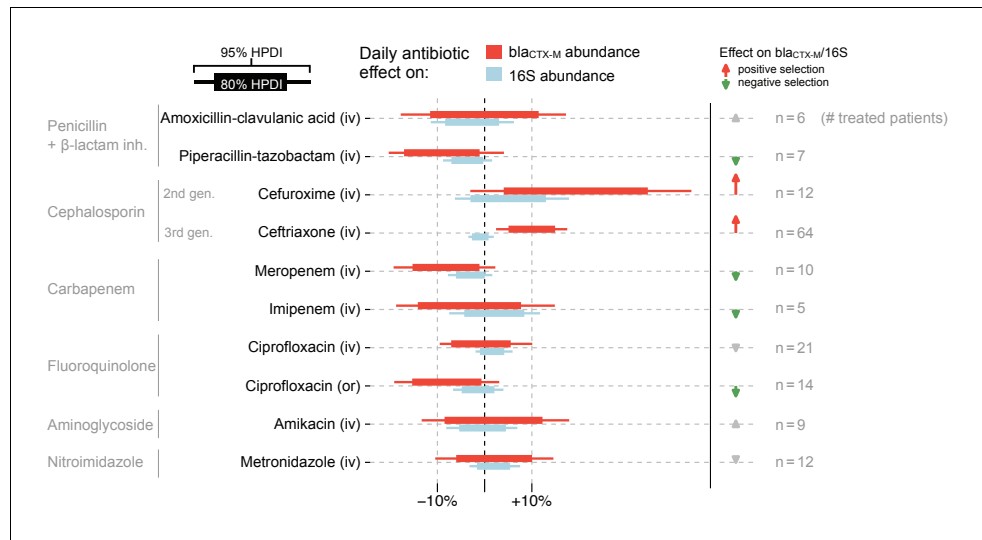

**Figure 4.** Estimated effects of different antibiotics on within-host dynamics from a multivariable model. The bars show estimated daily effects of individual antibiotics on the absolute bla$_{CTX-M}$ abundance (red) and 16S rRNA abundance (light blue) indicating the 80% and 95% highest posterior density intervals (thick and thin horizontal bars, respectively). The model also gives the antibiotic effect on the bla$_{CTX-M}$/16S relative resistance shown as arrows on the right-hand side. Arrows are in grey for antibiotics with mean effect estimates between −10% and +10%, otherwise they are coloured red (positive selection) and green (negative selection). Route of antibiotic administration (intravenous, iv; oral, or) is indicated in parenthesis.

The online version of this article includes the following figure supplement(s) for figure 4:

**Figure supplement 1.** Variability of replicate qPCR runs across the qPCR scale.
**Figure supplement 2.** Marginal posterior distributions for antibiotic effect parameters.
**Figure supplement 3.** Diagnostic plots of MCMC samples.

plus amikacin (by approximately 69% [80% CrI 20%, 89%] and 66% [80% CrI −7%, 88%], respectively) (*Figure 5*). Finally, a 14 day course of meropenem reduces bla$_{CTX-M}$ resistance carriage relative to a shorter 5 day course (by approximately 69% [80% CrI 20%, 89%]) (*Figure 5*).

## Discussion

By fitting a dynamic model accounting for both observation noise and within-host dynamics to time series data from 133 patients, we quantified the association between antibiotic exposure and changes in rectal swab abundance of gut bacteria and bla$_{CTX-M}$ resistance genes. The largest effects were found for exposures to the second and third generation cephalosporins, cefuroxime and ceftriaxone, both of which were associated with increases in bla$_{CTX-M}$ abundance. Forward simulations indicated that if these associations are causal, exposure to typical courses of these antibiotics would be expected to more than double the carriage duration of bla$_{CTX-M}$. Both cefuroxime and ceftriaxone have broad-spectrum killing activity (*Nahata and Barson, 1985*; *Neu and Fu, 1978*), but have limited activity against ESBL-producing organisms (*Livermore and Brown, 2001*; *Sorlózano et al., 2007*). Therefore, a direct selective effect of these two antibiotics is biologically plausible to account for the above finding.

Though credible intervals were wide, meropenem, piperacillin-tazobactam, and oral ciprofloxacin – all common agents for treating hospital-acquired infections (*Lautenbach et al., 2001*; *Masterton et al., 2003*; *Paterson, 2006*) – were associated with reductions in bla$_{CTX-M}$ abundance. All three are broad-spectrum antibiotics with activity against ESBL producers in the absence of specific co-resistance, suggesting that this association may at least in part be a causal effect. These antibiotics were also associated with a negative effect on relative resistance (bla$_{CTX-M}$ divided by 16S rRNA gene abundance). This observation can be explained by a general reduction of bacterial biomass that leads the bla$_{CTX-M}$ abundance to drop below detection levels. In line with this, our simulations suggested that a typical course of meropenem or of piperacillin-tazobactam would reduce

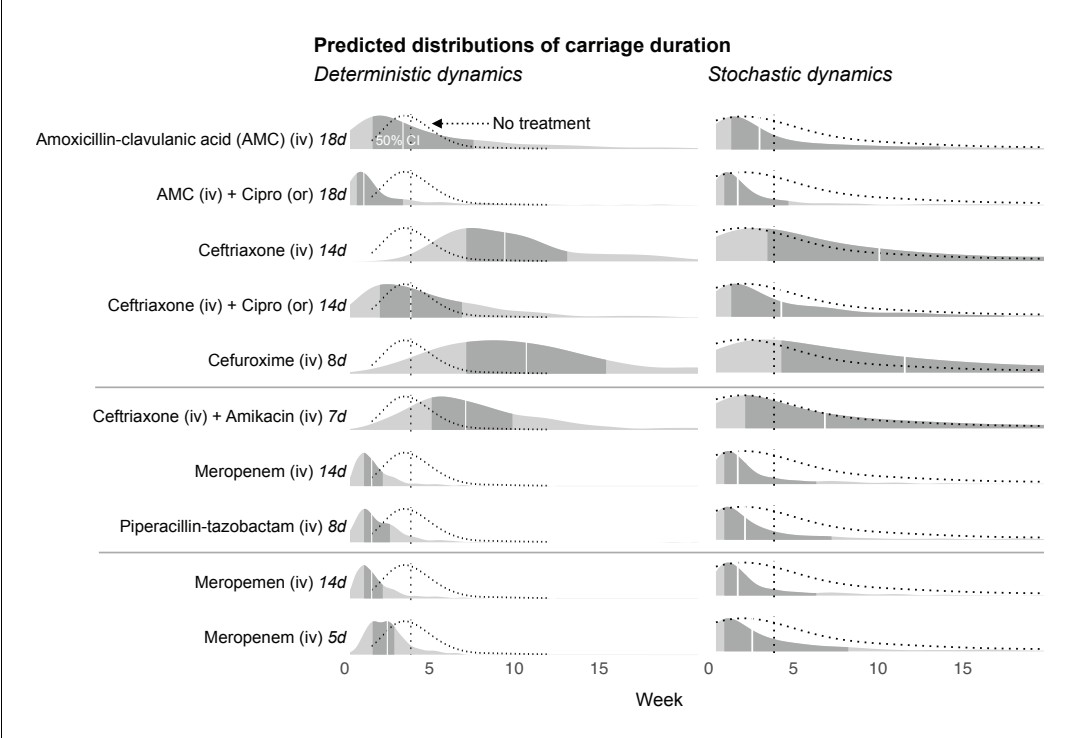

**Figure 5.** Simulated predictions of $bla_{CTX-M}$ carriage duration under different antibiotic treatments. The distributions on the left-hand side shows model predictions with parameter uncertainty, but assuming deterministic dynamics. The right-hand side shows the predictions with parameter uncertainty as well as Markov process uncertainty. The darker grey areas shows the 50% credible intervals and the white lines show the median predictions. Each density distribution is overlaid with the density line of the no treatment case (dotted line) and its median prediction (dotted vertical line). In the first five rows, we compare predictions for treatment with amoxicillin-clavulanic acid (18 days) and ceftriaxone (14 days), and each in combination treatment with ciprofloxacin. In the next three rows, we compare treatment with ceftriaxone plus amikacin (7 days), meropenem (14 days), and piperacillin-tazobactam (8 days). In the final two rows, we compare a 14 day course of meropenem with a shortened course (5 days). AMC stands for amoxicillin-clavulanic acid.

$bla_{CTX-M}$ carriage duration relative to no treatment by about 40%, and each course reduces $bla_{CTX-M}$ carriage duration by about 70% relative to a combined course of ceftriaxone plus amikacin. Pharmacokinetic models suggest that bacteriocidal serum concentrations of ceftriaxone persist for relatively short time periods after treatment (typically between 1 hr and 4d, *Garot et al., 2011*). In contrast, our model predicts that treatment effects on the gut flora can be much longer lasting (on the order of weeks). Also, a 14 day course of meropenem is predicted to reduce ESBL resistance carriage duration relative to a shortened course (5 days) by approximately 70%. While these findings suggest suppression of ESBL producing bacteria by use of carbapenem as a measure to reduce the risk of infection in high-risk patients, such a measure clearly needs to be balanced against the considerable risk of selecting for carbapenem resistance. Finally, we also found that adding oral ciprofloxacin to amoxicillin-clavulanic acid or ceftriaxone was associated with reductions in predicted median carriage duration of ESBL-producing bacteria by approximately 50%. This is in line with recent work indicating that hospital monotherapy with cephalosporins is more strongly association with later ESBL carriage relative to combination therapy (*Tacconelli et al., 2020*). Administration of oral fluoroquinolone to reduce faecal load of ESBL-producers in asymptomatic carriers has been used in outbreak settings with ESBL *E. coli* and *K. pneumoniae* (*Paterson et al., 2001*), where risk for fluoroquinolones resistance was low. However, the wide variation in rates of ciprofloxacin resistance amongst ESBL-producing *Enterobacteriaceae* across settings (*Winokur et al., 2001*) is likely to limit the generalisability of this finding. Although oral ciprofloxacin showed an association with reduced $bla_{CTX-M}$ abundance, intravenous ciprofloxacin showed near zero effect. Antibiotic selection for resistance due to antibiotics with different routes of administration has been previously explored in a mouse model, suggesting that, indeed, oral drug administration has stronger selective effect on resistance than intravenous administration (*Zhang et al., 2013*), but similar studies for humans are

lacking. Delineating the relationship between the various routes of antibiotic administration and resistance selection will be important for a better understanding of advantages and disadvantages of different routes of administration.

There are are number of advantages to our modelling approach over more classical, associational methods used in related work (*Meletiadis et al., 2017*). First, because we use a mechanistic model it allows us to directly estimate ecologically important parameters such as strength of resistance amplification under antibiotic selection, which are of inherent interest and can inform further modelling work. Indeed, our predictions of resistance carriage duration (*Figure 5*) are good examples of the latter. Second, our model uses the variability present in the data to quantify different types of data variability due to the data collection (noise from the qPCR machine, noise from taking the swab and DNA extraction). This allows our model to fully propagate uncertainty to the final estimates. Finally, rather than using only aggregate data (which loses information), our analysis is designed to fully exploit the information available in the time series data. Our work also has a number of important limitations, aside from the obvious risks of confounding present in this observational dataset. Our analysis did not explicitly model changing antibiotic concentrations in the gut, nor did it attempt to explicitly model how antibiotics affect the ecology of the gut bacterial community. While it would have been straightforward to include a pharmacokinetic model of antibiotic concentrations (similar studies have been performed in mice [*Jumbe et al., 2003*] and pigs [*Nguyen et al., 2014*]), disentangling direct effects of antibiotic concentrations from indirect effects mediated by other components of the gut flora is far more challenging and beyond the scope of what we considered appropriate with the available data. Instead, our model assumed multiplicative antibiotic effects, which we considered a reasonable simplification of the underlying mechanisms. Multiplicative effects imply that antibiotics alter the daily total bacterial growth rate (16S abundance) and the growth of resistant bacteria relative to the average bacterium ($bla_{CTX-M}$/16S), but it does not allow for more complex fitness effects due to, for example, synergies between antibiotics (*MacLean et al., 2010*), or density-dependent effects, whereby antibiotic-mediated killing may depend on bacterial density (*Udekwu et al., 2009*). Further, many non-antibiotic drugs have been shown to have an impact on human gut bacteria (*Maier et al., 2018*), but only antibacterials were considered in our analysis. Lastly, all patients in this study were identified (and consenting) ESBL-carriers. Therefore, apart from the potential for selection bias, we assumed that all changes in $bla_{CTX-M}$ abundance were due to within-host dynamics, neglecting the possibility of new acquisitions, which should be the scope for other modelling frameworks that integrate both within- and between-host dynamics.

Antibiotic impact on the human gut microbiome is likely to be an important mediator for the increase of bacterial resistance globally (*Donskey, 2004*; *Relman and Lipsitch, 2018*). A large body of theory has been developed that demonstrates the role of within-host processes for understanding population-wide selection of resistance through antibiotic use (*Blanquart, 2019*; *Davies et al., 2019*; *Webb et al., 2005*; *Knight et al., 2018*; *Lipsitch et al., 2000*; *Lipsitch and Samore, 2002*; *Webb et al., 2005*). However, surprisingly few studies have used data to quantify the effect of antibiotic treatment on resistance abundance within individual gut microbiomes. Two studies involving patients admitted to intensive care units looked at the effect of a preventative antibiotic cocktail (selective digestive decontamination) on gut microbiome resistance in patients, with one study (including n = 13 patients) finding no clear effect (*Buelow et al., 2014*), and the other (n = 10) showing increases of four different resistance genes associated with treatment (*Buelow et al., 2017*). *Gibson et al. (2016)* studied the faecal metagenomics of preterm infants (n = 84) over time and found that treatment with antibiotics was correlated with enrichment of both cognate and noncognate resistance markers. But none of these studies attempted to model resistance dynamics. The modelling framework we have developed enables testable predictions about the impact of different antibiotics and lengths of treatment on the duration of carriage of resistant determinants above detection threshold. Such understanding is an important step toward understanding the spread of antibiotic resistance.

The development of a mechanistic understanding of the relationship between antibiotic use in a population and the proportion of this population in whom resistance can be detected relies on quantifying the antibiotic effects in individual exposed patients, as we do here, but also on quantifying the knock-on effects on transmission to contacts. These indirect effects are likely to be considerable. A recent study in Dutch travellers returning to the Netherlands who had acquired ESBL-producing *Enterobacteriaceae* carriage overseas found that their new carriage status was associated with a

150% increase in the daily risk of non-carrying household members also becoming ESBL-positive (*Arcilla et al., 2017*). Developing mechanistic models for the spread of ESBLs and other resistance determinants within host populations accounting for direct and indirect antibiotic effects is an important priority for future research. Such models would help us to understand and predict how changes in antibiotic usage patterns affect the prevalence of antimicrobial resistance in a community and ultimately help to prioritise interventions to reduce the burden of antimicrobial resistance.

## Materials and methods

### Study participants and follow-up

Participants were recruited as part of an observational, prospective, cohort study that included three hospitals (Italy, Serbia and Romania), with known high endemic prevalence of antibiotic resistance in bacterial infections. The hospitals were serving a general urban population. The study was conducted over two years from January 2011 to December 2012 as part of the multi-centre SATURN ('Impact of **S**pecific **A**ntibiotic **T**herapies on the prevalence of h**U**man host **R**esista**N**t bacteria') project (NO241796; clinicaltrials.gov NTC01208519). The study enrolled adult (>18 y) inpatients of medical and surgical wards, excluding pregnant patients. Enrolled patients were screened at admission for carriage of ESBL-producing *Enterobacteriaceae* with rectal swabs (E swab, Copan, Italy). Patients who tested positive for ESBL producing *Enterobacteriaceae* carriage (details below) were included in the follow-up cohort. The target cohort size was calculated to be 400 patients, based on the number of patients that would be required to detect one log difference in resistance abundance with 90% certainty. However, patient recruitment we slower than anticipated. For all follow-up patients (n = 133) rectal swabs were taken every two to three days (as per study protocol) during hospitalisation, which includes one swab at admission and one at discharge. The swabs were stored at −80 degrees Celsius and sent to a central laboratory for processing. Using patient charts, the study also collected information on antibiotics treatment, including antibiotic type, duration, and route of administration. See *Table 1* for an overview of the study details. Written informed consent and consent to publish was obtained from all patients before study enrolment. All collected data was entered de-identified into the central study database which sat in Tel Aviv in accordance with the local rules of personal data privacy protection. The study protocol was reviewed and approved by the Catholic University Ethics Commission in Rome, Italy (protocol P/291/CE/2010 approved on 6.4.2010) and the Clinical Center of Serbia Ethic Committee (protocol 451/34 approved on 18.03.2010). At the site in Romania patient screening for multidrug-resistant bacteria was considered, due to the local epidemiology, a quality improvement intervention and did not require institutional ethical approval.

### Identification of ESBL producing organism carriers

Samples taken at admission were cultured on chromogenic agar (Brilliance ESBL, Oxoid, Basingstoke, UK) to test for ESBL producing *Enterobacteriaceae*. Characteristically coloured colonies for *Enterobacteriaceae* were isolated (single colony per colour), replated on blood agar and incubated overnight in air. ESBL status was then confirmed with the double disk diffusion method according to CLSI guidelines . These methods were performed in the laboratories on the respective hospitals (all laboratories were ISO accredited). The above methods are not specific to any single bacterial species, but instead identify ESBL producing specimens of the *Enterobacteriaceae* family, including *Escherichia coli*, *Klebsiella pneumoniae*, or *Enterobacter cloacae* and others. According to the standard definition by the Centers of Disease Control (*CDC/NHSN, 2018*), samples taken at admission identify community acquired organisms.

### Quantitative PCR

DNA was extracted from the swab samples using QIAampDNA Stool Mini Kit (Qiagen) and a fixed volume (4 μl) of DNA solution was used as a template for quantitative PCR (qPCR) assays. Two singleplex qPCR assays were conducted, one to assess quantity of $bla_{CTX-M}$ gene family with primers CTX-M-A6 (TGGTRAYRTGGMTBAARGGCA) and CTX-M-A8 (TGGGTRAARTARGTSACCAGAA) (product length, 175 bp) and one targeting a conserved bacterial 16S rRNA gene region bacteria using the following primer set, 16S_E939F (GAATTGACGGGGGCCCGCACAAG) and 16S_1492R

(TACGGYTACCTTGTTACGACTT) (product length, 597 bp) to assess total bacterial quantity. Each singleplex qPCR run targeted either 16S gene or $bla_{CTX-M}$, and included reaction tubes with negative controls and tubes containing a standard (16S or $bla_{CTX-M}$ depending on the run) of different concentrations (eight different dilutions), which also served as a positive control. For details on reaction mix and cycling conditions see *Lerner et al., 2013*. All reactions were carried out in duplicates, sometimes triplicates, representing technical replicates. The qPCR was performed at the Laboratory of Medical Microbiology, University of Antwerp (ISO accredited). While we did not validate the 16S qPCR measurements through, for example, spiked standard DNA preparations, the ability of qPCR to quantify bacterial amounts in faecal samples has been shown previously (*Rinttilä et al., 2004*).

## Time series autocorrelation

We first transformed all qPCR measurements onto a log-scale. For all patients and each time point we then computed the mean of the qPCR duplicates (or triplicates) for $bla_{CTX-M}$ and 16S rRNA. To get reliable estimates of autocorrelation, we selected only patients with more than five time points. Separately for the $bla_{CTX-M}$ and 16S rRNA gene data, we computed the first-order autocorrelation (disregarding varying spacing between time points) for each patient, and we averaged these values across the patients. We then simulated serially uncorrelated 'white noise' time series, again separately for $bla_{CTX-M}$ and 16S rRNA, with the same length as the patient data and with identical time series mean and variance. Similar to the real data, we computed mean autocorrelations for the simulated data and show their distribution for a large number of such simulations (n = 10,000) together with the observed autocorrelation (*Figure 1—figure supplement 1a and b*). We also computed the proportion of simulated datasets that showed an average autocorrelation equal to, or larger than, the observed data, and we show those numbers on the arrows in *Figure 1—figure supplement 1a and b*.

## Estimating observation and process noise in time series

To estimate the amount of observation noise and process noise in the time series we constructed a Bayesian state-space model that included qPCR noise, swab noise, and biological noise. This model is given through:

$$
\begin{aligned}
q_{i,j,k,g} &\sim N(s_{i,j,g}, \sigma_{qpcr}), \\
s_{i,j,g} &\sim N(x_{i,j,g}, \sigma_{swab_g}), \\
x_{i,j+1,g} &\sim N(s_{i,j,g}, \sigma_{bio_g}),
\end{aligned}
\tag{1}
$$

where $i$ denotes a given patient, $j$ denotes a swab (one per time point), $k$ denotes a qPCR measurement (multiple repeats per swab), and $g$ denotes the genetic target, either $bla_{CTX-M}$ or 16S rRNA. The term $q_{i,j,k,g}$, represents the measured quantity of genetic target $g$, of the kth qPCR replicate (on a log-scale) from patient $i$, at time point $j$. In addition, there are two hidden-state parameter vectors: $s_{i,j,g}$ is the underlying, true sequence abundance of genetic target $g$ that a qPCR assay with 100% efficiency could (in theory) measure at time point $j$ for patient $i$, and $x_{i,j,g}$ is the actual gene abundance of genetic target $g$, in the patient at time point $j$ for patient $i$, before the added noise through the swab process and gene extraction. The unobserved variables of interest are $\sigma_{qpcr}$, the qPCR machine error (assumed to be the same for $bla_{CTX-M}$ and 16S rRNA), $\sigma_{swab_g}$, the swab variation of the genetic target $g$, and $\sigma_{bio_g}$, the variation of genetic target $g$ from biological processes. We assigned improper uniform ('flat') priors for the hidden-state parameters and generic weakly informative priors (half-normal, $N^+(0,1)$) for the the noise parameters $\sigma_{qpcr}$, $\sigma_{swab_g}$, and $\sigma_{bio_g}$. We then fitted this model to the $bla_{CTX-M}$ and 16S rRNA measurements. The posteriors of the three noise parameters are shown in *Figure 1—figure supplement 1c*, where we expressed each type of noise as a fraction of the total noise. The model was fitted using Stan software (v2.19.1) (*Carpenter et al., 2017*) and with additional analysis in R (*R Development Core Team, 2016*), and we sampled 80,000 samples from the posterior with four independent chains and a burn-in period of 20,000 samples. We assessed convergence by checking that R̂ (*Gelman and Rubin, 1992*) was low (<1.01) for all parameters, and visually by looking at the rank plots for all parameters (shown in *Figure 2—figure supplement 1a*). For rank plots, posterior draws are ranked across all chains and then ranks are plotted as histograms separately for each chain. A uniform shape of all histograms then indicates that all chains target the same posterior (*Vehtari et al., 2019*).

## Between and within time series variance

For the estimation of between and within time series variation we used a Bayesian hierarchical model, which accounted for unbalanced sampling between patients. This model used the mean posterior estimates of $x_{i,j}$ (actual gene abundance in time point $j$ for patient $i$) from the previous model, and it took the form

$$\begin{aligned} x_{i,j,g} &\sim N(\mu_{i,g}, \sigma_{within,g}), \\ \mu_{i,g} &\sim N(\mu_g, \sigma_{between,g}), \end{aligned} \qquad (2)$$

where $mu_{i,g}$ is the mean abundance of genetic target $g$ (bla$_{CTX-M}$ or 16S gene) and patient $i$, around which the log-scaled measurements were assumed to be normally distributed with standard deviation $\sigma_{within,g}$, the within time series variation. The mean abundances were assumed to follow a normal distribution with a population mean μ and between-patient standard deviation $\sigma_{between}$. We assigned improper uniform priors for the population and the patient means, and generic weakly informative priors for the standard deviations ($N^+(0,1)$). We fitted the model using Stan (v2.19.1), with 80,000 posterior draws after a burn-in period of 500 iterations. The $\hat{R}$ statistic (<1.01) and MCMC rank histogram plots (*Figure 2—figure supplement 1b*) were used to assess convergence. Model estimates are shown in *Figure 2*. To calculate the coefficient of variation for the non-log-scaled bla$_{CTX-M}$ and 16S rRNA measurements, we use the transform described by *Koopmans et al., 1964*:

$$c_v = \sqrt{e^{s_{ln}^2} - 1}, \qquad (3)$$

where $s_{ln}$ is the estimated standard deviation of the log-scaled data.

## Association of antibiotic treatment and changes in resistance

To study the association between antibiotic treatment and resistance we looked at relative abundance of resistance (bla$_{CTX-M}$ abundance/16S rRNA gene abundance) as a marker of natural selection. First, we computed the changes in relative resistance for every pair of adjacent time points and for each antibiotic we used a binary variable indicating whether or not a given antibiotic was administered between these time points. When an antibiotic treatment was on the same day as a swab, this treatment was allocated to the time interval between this day and the next swab. We first looked at how changes in relative resistance are associated with courses of any antibiotics, then with courses of anti-enterbacteriaceae antibiotics to which carriage of bla$_{CTX-M}$ does not confer direct resistance (colistin, meropenem, ertapenem, imipenem, amoxicillin-clavulanic acid, ampicillin-sulbactam, piperacillin-tazobactam, gentamicin, amikacin, ciprofloxacin, ofloxacin, levofloxacin, tigecycline, doxycycline), and finally with antibiotics that have a broad-spectrum activity and to which bla$_{CTX-M}$ does confer resistance (cefepime, ceftazidime, ceftriaxone, cefotaxime, cefuroxime, amoxicillin, ampicillin). Results are shown in *Figure 3*, upper panel. We evaluated how likely the observed differences between treatments are under the assumption of no association between treatment and resistance. For this we did a permutation or 'reshuffling' experiment: we randomly reassigned (without replacement) the antibiotic treatment labels to the data intervals. We compute the distribution of mean differences from 50,000 permutations and compare this to the observed difference (*Figure 3*, lower panel).

## Dynamic within-host model

We extended previous modelling approaches to extracting ecological parameters from microbial ecosystem dynamics (*Faust and Raes, 2012*; *Stein et al., 2013*) by applying a Bayesian hidden-state model, which featured two layers of hidden-state variables: the unobserved mean of the qPCR measurements, and the unobserved true abundances of bla$_{CTX-M}$ or 16S rRNA in the gut. This model structure allowed us to separate process noise (stochastic effects impacting the gene abundance change from one day to the next) from observation noise (stochastic effects impacting the swab efficiency, DNA extraction or the qPCR measurements) and also to account for different spacing between measured time points.

We analysed antibiotic treatment separately by type and route of administration, only including treatments that occurred in five or more patients (amoxicillin-clavulanic acid (iv), piperacillin-tazobactam (iv), cefuroxime (iv), ceftriaxone (iv), meropenem (iv), imipenem (iv), ciprofloxacin (iv),

ciprofloxacin (or), amikacin (iv), metronidazole (iv)). Since the $bla_{CTX-M}$ and 16S rRNA measurements are expected to be proportional to the absolute abundance of the resistance gene and bacterial load respectively, the ratio $bla_{CTX-M}$/16S is a measure of the relative abundance of the $bla_{CTX-M}$ gene in the gut microbiota. Positive or negative selective effects by antibiotics on $bla_{CTX-M}$ mediated resistance are expected to cause shifts in bacteria carrying $bla_{CTX-M}$ versus non-carriers. As a result they affect $bla_{CTX-M}$/16S, but quantifying their effects on absolute $bla_{CTX-M}$ abundance is important for predicting extinction and persistence of the $bla_{CTX-M}$ gene. Under the assumption that 16S rRNA gene abundance is independent of antibiotic treatment, variation in 16S rRNA would be caused mainly by the swab procedure and DNA extraction (and other steps in the protocol), and it could be used to normalise $bla_{CTX-M}$ abundance. However, as we found in *Figure 4*, certain antibiotic treatments were associated with changes in 16S rRNA abundance. Thus, we used a dynamic model that explicitly modelled both antibiotic effects on 16S rRNA and on $bla_{CTX-M}$/16S, from which the effects on $bla_{CTX-M}$ could then be computed.

Studying the standard deviation between qPCR measurement repeats as a function of the mean, we observed that qPCR variation remained relatively stable over five orders of magnitude of the mean measurement (from 1.5 to 6.5 on the log-scale), but it increased quickly for lower magnitudes (*Figure 4—figure supplement 1*). In the Bayesian model for different sources of variation described above, the parameter $\sigma_{qpcr}$ assumed that the qPCR uncertainty is the same across measurements. Here, we aimed to account for the fact that low measurements of gene copy numbers have higher uncertainty. We fitted a smooth spline (choosing five degrees of freedom) to the qPCR measurements (red line in *Figure 4—figure supplement 1*). This let us assign an estimated qPCR standard deviation to every set of qPCR repeats. We provided those estimates as data to the Bayesian model. This allowed us to use all qPCR measurements, including extremely low values, without removing any data points from the analysis. Our model then took the form:

$$
\begin{aligned}
q_{i,j,g,k} \quad &\sim N(s_{i,j,g}, \sigma_{qpcr_{i,j,g}}), \\
s_{i,j,g=16S} \quad &\sim N(x_{i,j,b=16S}, \sigma_{swab}), \\
x_{i,j,b=ratio} \quad &= s_{i,j,g=CTX-M} - s_{i,j,g=16S}, \\
x_{i,j+1,b} \quad &\sim N(x_{i,j,b} + f(abx)_{i,j,b}, \sqrt{t_{j+1} - t_j}\, \sigma_{bio,b}), \\
f(abx)_{i,j,b} \quad &= \sum_{t_j}^{t_{j+1}-1} \left( a_b + \sum_{z=1}^{n_z} c_{z,b} y_{z,t} \right),
\end{aligned} \tag{4}
$$

where $g$ denotes either $bla_{CTX-M}$ or 16S rRNA, and $b$ denotes either relative resistance ($bla_{CTX-M}$/16S rRNA) or 16S rRNA. Then, $q_{i,j,g,k}$ is the $k$-th qPCR result (log-scaled) of patient $i$, measured in the sample with index $j$, and genetic target $g$ ($bla_{CTX-M}$ or 16S rRNA). The qPCR standard deviation for sample $j$ and patient $i$ is given through $\sigma_{qpcr_{i,j,g}}$, and it is estimated as described above. $s_{i,j,g}$ is the mean of the qPCR measurements in sample $j$ of patient $i$, and genetic target $g$, and $x_{i,j,b=16S}$ is the 16S sequence abundance that is actually present at time point $j$ in the gut of patient $i$. The error introduced by the swab procedure, DNA extraction etc. is given through $\sigma_{swab}$, and we assume that this error causes the same perturbations to $bla_{CTX-M}$ and to 16S, such that this error cancels out when computing the ratio of $bla_{CTX-M}$ and 16S rRNA $x_{i,j,b=ratio}$, which on a log-scale is computed as the difference ($s_{i,j,g=CTX-M} - s_{i,j,g=16S}$). Further, $t_j$ denotes the calendar day of sample $j$ in patient $i$, and $t_{j+1}$ denotes the calendar day of the following sample in the same patient $i$. The daily biological variability of the $bla_{CTX-M}$/16S ratio and of 16S are given through $\sigma_{bio,b=ratio}$ and $\sigma_{bio,b=16S}$, respectively, with $\sqrt{t_{j+1} - t_j}$ adjusting the expected random walk variation by the number of days between observations (*Lemons and Langevin, 2002*). The ecological dynamics are modelled with the function $f(abx)_{i,j,b}$, which is the change in the expected value of $x_{i,j,b}$ between sample $j$ and $j+1$. In the definition of $f(abx)_{i,j,b}$ in line 5 of *Equation 4*, $a_b$ denotes the neutral growth or loss of $g$, $c_{z,b}$ denotes the effect of antibiotic $z$ on $b$, and $y_{z,t}$ is a boolean variable indicating whether or not antibiotic $z$ was given on day $t$. The term inside the bracket takes, for a single calendar day $t$, the neutral growth/loss term ($a_b$) and adds to this the summed effect of all antibiotics given on day $t$. This is computed for each calendar day from $t_j$ until a day before the subsequent sample ($t_{j+1} - 1$). The effects of all of these days are then summed up. Note, that $x_{i,j,b}$ denotes the abundance of 16S rRNA, or the relative abundance of $bla_{CTX-M}$/16S rRNA, on the log-scale. Exponentiating this variable gives the copy numbers (or copy number ratio) on the real scale. Therefore, summing all effects on the scale of $x_{i,j,b}$ is equivalent to

multiplying the exponentiated effects on the scale of copy numbers. For example, consider that genetic target $b = 16S$ in patient $i = 1$ has at the time of sample $j = 1$ an abundance of $10^{x_{i=1,j=1,b=16S}} = 100$ copy numbers and a neutral trend of $a_{b=16S} = -0.5$. Suppose on the day of this sample ($t_{i=1,j=1}$) two antibiotics $z = 1$ and $z = 2$ are given with effects $c_{z=1,b=16S} = +0.5$ and $c_{z=2,b=16S} = +0.1$, then one day after this sample the genetic target abundance has an expected abundance of $100 * 10^{-0.5} * 10^{+0.5} * 10^{+0.1} = 126$.

In this model, $q_{i,j,g,k}$ and $y_{z,t}$ correspond to measured data, $\sigma_{qpcr_{i,j,g}}$ is computed from this data (see above). All other parameters are estimated: the hidden-state variables are $s_{i,j,g}$, $x_{i,j,g=16S}$, and $x_{i,j,g=ratio}$, the noise variables are $\sigma_{swab}$, $\sigma_{bio,g=16S}$, and $\sigma_{bio,g=ratio}$, and finally the variables describing the ecology are $a_g$, $c_{z,g}$. The model has three likelihood functions. The first (line 1) applies to each single qPCR result and it relates repeat qPCR measurements to their variability and underlying mean. The second (line 2) relates qPCR means of 16S rRNA to their variability and the underlying 16S rRNA gene abundance. The third likelihood (line 4) applies to all sample pairs where a previous and following sample exists from the same patient. This likelihood relates changes in underlying 16S rRNA abundance or underlying bla$_{CTX-M}$/16S rRNA to the parameters $a_b$ and $c_{z,b}$, and $\sigma_{bio}$. All parameters (including all hidden-states) are estimated simultaneously. We can express the posterior distribution over all estimated parameters ($\Theta$), which is conditional on the set of all data ($D$), and the prior over the parameter space $p(\Theta)$. For readability, here we only keep necessary subscripts:

$$
\begin{aligned}
p(\Theta|D) &= p(s, x, a, c, \sigma_{swab}, \sigma_{bio} | q, \sigma_{qpcr}, y), \\
p(\Theta|D) &\propto p(\Theta) \prod_{i,j,g,k} p(q_{i,j,g,k} | s_{i,j,g}, \sigma_{qpcr_{i,j,g}}) \prod_{i,j} p(s_{i,j,g=16S} | x_{i,j,b=16S}, \sigma_{swab}) \\
&\qquad \prod_{i,j,b} p(x_{i,j+1,b} | c_b, a_b, \sigma_{bio,b}, x_{i,j,b}, y).
\end{aligned}
$$

(5)

On the hidden-state variables we assigned improper uniform priors, on the standard deviations describing swab and biological variability we assigned standard weakly informative priors ($N^+(0,1)$), and on the antibiotic effects ($c$) we assigned conservative priors of the form N(0, 0.1). We fitted the model using Stan software (v2.19.1), and we sampled 80,000 samples from the posterior with four independent chains after a burn-in phase of 10,000 samples. The marginal posterior draws of the $c_{z,g}$ parameters are exponentiated to be on the scale of gene counts. Subtracting one allows us to express daily effects as percent of increase or decrease relative to the previous day (*Figure 4*). We also show marginal posterior distributions together with prior distributions for the $c_{z,g}$ parameters (*Figure 4—figure supplement 2*). Diagnostic plots for the MCMC sampling of the $c_{z,g}$ parameters are shown in *Figure 4—figure supplement 3*.

We compared our model as given through model definition four to a model without antibiotic effects (all $c$ parameters set to zero). The number of patients treated with the same antibiotic is too small to perform cross-validation (iteratively fitting the model to all time series while leaving out data from one patient, which is then used to assess model predictions). Therefore, we used an efficient approximation of Bayesian leave-one-out cross-validation using Pareto-smoothed importance sampling (*Vehtari et al., 2017*).

We forward simulated bla$_{CTX-M}$ data using the dynamic model above and the posterior distributions from the model fit. We added to the model a threshold below which the bla$_{CTX-M}$ gene becomes extinct or at least undetectable. According to a study of returning European travelers to Southeast Asia, ESBL carriers lose detectable resistant bacteria after a median of 30 days (*Arcilla et al., 2017*). Accordingly, we simulated bla$_{CTX-M}$ time series without antibiotic treatment and chose an extinction threshold (0.25 bla$_{CTX-M}$ copy numbers) that achieved the same median extinction time. We then used this model to repeatedly (2000 times) simulate bla$_{CTX-M}$ carriage durations, with each simulation using a new draw from the parameter posterior. The resulting distribution of carriage times contains both the uncertainty in the parameter estimates and uncertainty from the Markov process (*Figure 5*, right-hand side). We also draw a set of parameter values from the posterior to simulate bla$_{CTX-M}$ carriage durations repeatedly (200 times) with the same parameters, and taking the median carriage duration to remove Markov process uncertainty. We repeated this for 300 draws of parameters (*Figure 5*, left-hand side). We used both of the above methods to simulate carriage time under different alternative antibiotic treatments. The resulting distributions are shown in *Figure 5*.

## Acknowledgements

We thank Jonas Schluter, Marc Lipsitch, Thomas Crellen, and Pierre Jacob for valuable feedback along the way. RN as well as the study were supported by funding from the European Community's R-GNOSIS Integrated project (FP7/2007-2013) under grant agreement number 241796. RN and BSC were also supported by the Medical Research Council and Department for International Development (grant number MR/K006924/1). BSC works within the Wellcome Trust Major Overseas Programme in SE Asia (grant number 106698/Z/14/Z). The funders had no role in study design, data collection and interpretation, or the decision to submit the work for publication.

## Additional information

### Competing interests

Ben S Cooper: Reviewing Editor, eLife. The other authors declare that no competing interests exist.

### Funding

| Funder | Grant reference number | Author |
| --- | --- | --- |
| Medical Research Council | | Rene Niehus<br>Ben S Cooper |
| Wellcome Trust | Major Overseas Programme, 106698/Z/14/Z | Ben S Cooper |
| European Union 7th Framework Programme | R-GNOSIS Integrated project, grant agreement number 241796 | Rene Niehus<br>Esther van Kleef<br>Agata Turlej-Rogacka<br>Christine Lammens<br>Yehuda Carmeli<br>Herman Goossens<br>Evelina Tacconelli<br>Biljana Carevic<br>Surbhi Malhotra-Kumar<br>Ben S Cooper |
| Department for International Development | MR/K006924/1 | Rene Niehus<br>Ben S Cooper |
| National Medical Research Council | | Yin Mo |

The funders had no role in study design, data collection and interpretation, or the decision to submit the work for publication.

### Author contributions

Rene Niehus, Conceptualization, Data curation, Software, Formal analysis, Validation, Investigation, Visualization, Methodology; Esther van Kleef, Conceptualization, Resources, Data curation, Formal analysis, Validation, Investigation, Visualization, Methodology, Project administration; Yin Mo, Conceptualization, Validation, Methodology; Agata Turlej-Rogacka, Resources, Validation, Investigation, Methodology; Christine Lammens, Resources, Data curation, Validation, Methodology; Yehuda Carmeli, Surbhi Malhotra-Kumar, Conceptualization, Resources, Supervision, Funding acquisition, Validation, Investigation, Methodology, Project administration; Herman Goossens, Conceptualization, Supervision, Funding acquisition, Validation, Investigation, Methodology, Project administration; Evelina Tacconelli, Conceptualization, Supervision, Validation, Investigation, Methodology, Project administration; Biljana Carevic, Resources, Investigation, Methodology; Liliana Preotescu, Resources, Investigation; Ben S Cooper, Conceptualization, Formal analysis, Supervision, Funding acquisition, Validation, Investigation, Methodology, Project administration

### Author ORCIDs

Rene Niehus https://orcid.org/0000-0002-6751-4124
Ben S Cooper http://orcid.org/0000-0002-9445-7217

## Ethics

Clinical trial registration NCT01208519.

Human subjects: Written informed consent and consent to publish was obtained from all patients before study enrolment. All collected data was entered de-identified into the central study database which sat in Tel Aviv in accordance with the local rules of personal data privacy protection. The study protocol was reviewed and approved by the Catholic University Ethics Commission in Rome, Italy (protocol P/291/CE/2010 approved on 6.4.2010) and the Clinical Center of Serbia Ethic Committee (protocol 451/34 approved on 18.03.2010). At the site in Romania patient screening for multidrug-resistant bacteria was considered, due to the local epidemiology, a quality improvement intervention and did not require institutional ethical approval.

## Decision letter and Author response

Decision letter https://doi.org/10.7554/eLife.49206.sa1
Author response https://doi.org/10.7554/eLife.49206.sa2

## Additional files

### Supplementary files

- Transparent reporting form

- Reporting standard 1. STROBE guidelines.

- Reporting standard 2. STROBE AMS guidelines.

- Reporting standard 3. MICRO guidelines.

### Data availability

The Bayesian model code in Stan, the R code to fit the model, and the data in Rdata format as well as csv format are deposited in Dryad under https://doi.org/10.5061/dryad.8vf034t.

The following dataset was generated:

| Author(s) | Year | Dataset title | Dataset URL | Database and Identifier |
|---|---|---|---|---|
| Niehus R, van Kleef E, Yin M, Turlej-Rogacka A, Lammens C, Carmeli Y, Goossens H, Tacconelli E, Carevic B, Malhotra-Kumar S, Cooper BS | 2020 | Quantifying antibiotic impact on within-host dynamics of extended-spectrum beta-lactamase resistance in hospitalized patients | https://doi.org/10.5061/dryad.8vf034t | Dryad Digital Repository, 10.5061/dryad.8vf034t |

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
