## [Decision Letter]

**Acceptance summary:**

The dynamics of resistance in response to antibiotic treatment is of central importance in understanding resistance emergence more broadly. This paper develops a data-driven model to describe the within-host dynamics of *Enterobacteriaceae* that produce extended-spectrum β-lactamase. Supported by the model, the authors compare different antibiotics' effects on resistance.

**Decision letter after peer review:**

Thank you for submitting your article "Quantifying antibiotic impact on within-patient dynamics of extended-spectrum β-lactamase resistance" for consideration by *eLife*. Your article has been reviewed by three peer reviewers, and the evaluation has been overseen by a Reviewing Editor and Wendy Garrett as the Senior Editor. The following individuals involved in review of your submission have agreed to reveal their identity: Amy Mathers (Reviewer #2); Lulla Opatowski (Reviewer #3).

The reviewers have discussed the reviews with one another and the Reviewing Editor has drafted this decision to help you prepare a revised submission.

Summary:

This work asks what the dynamic consequences of antibiotic perturbations are to the human gut flora, and addresses this question with a novel mathematical model that is fit to data. The authors use the model to estimate the impact of antibiotic treatment on the abundance of resistance (through bla_CTX-M_) and on the duration of carriage of resistance. Overall, the manuscript represents a potentially important contribution to the field. However, there is some T data overlap in a previous study by the authors. and we have identified a few essential revisions to resolve this issue and that need to be addressed as well.

Essential revisions:

The reviewers brought up the previous publication, Meletiadis, J. et al., 2017, in which a good portion of the cases were reported, and where some of the same conclusions were reached.

The authors should clarify the novel contributions of this work in light of the previous manuscript. It seems that the key claimed results here are (1) variability, Figures 1, 2; (2) effect of treatment on relative resistance; (3) effect of treatment on abundance and (4) the model with fitting, and consequently the estimates re duration of carriage. Each has limitations.

Please set these results, with the novelty and limitations, in context.

The modelling is clearly new and I think it is important. In that direction, the clarity should be improved. I particularly noted the very long paragraph in subsection “Dynamic within-host model”, and the benefit that explaining the model and its assumptions (and limitations) for non-statisticians, non-modellers would likely bring.

Please clarify re interpretation of 16s, abundance as noted by reviewers.

Please improve Figures 1 and 2. While "hieroglyphics" may go a bit far, I agree that Figure 1 is visually striking but hard to see and interpret. Perhaps in Figure 1 you could group the time curves by similarity, or present an expanded view of some representative curves, with the full set in an appendix. Or something like that.

Reviewer #1:

The motivation for this study and its goals are certainly important and timely. The collateral effects of antibiotic treatment on the distribution of bacteria species and strains in the enteric flora (microbiome) of humans and the incidence of antibiotic resistance genes and resistance encoding plasmids are subjects of considerable importance, epidemiologically as well as clinically. In their Introduction, the authors do a fine job of presenting this and particularly so for the extended-spectrum β-lactamase (ESBL)-producing *Enterobacteriaceae* and the blaCTX family of resistance genes, that are the focus of their study.

This investigation certainly has the virtue of being extensive; fecal samples were taken sequentially from 133 hospitalized patients from three countries (Romania, Serbia and Italy) for a median of five samples from each patient. This virtue does have a downside; these patients were hospitalized for different reasons and treated orally of intravenously with 10 different antibiotics. The results presented suggest that some antibiotics are more likely than others to affect the distribution and abundance of these enteric bacteria and the bla_CRX-M_ genes, their study is correlative rather than mechanistic. From the results presented, it's not clear how the different modes and purpose of treatment the within-patient dynamics ESBL resistance. From what they say in their Discussion, the authors are very much aware of this limitation of this study.

Although their observation some antibiotics are more likely than others to affect the enteric microbiome and the frequency of bla_CRX-M_ genes could have epidemiological and clinical implication, they don't really consider these potential implications of their results. It's also not clear how this information can be used. Presumably the choice of antibiotics, mode and frequency of administration in these hospitalized patients is based on the nature of the infection and information about the susceptibility of the target bacteria to the drugs employed for treatment. While in an ideal world, consideration should to the collateral effects of treatment; we are far from that ideal world. This is particularly so for hospitalized patients. It would be of some interest to do an analogous study for the treatment of community-acquired infections (about 90% of antibiotic use in humans), where information about the collateral effects of treatment with different antibiotics may have broader and implications that could be implemented.

Reviewer #2:

This is an interesting and important piece of work demonstrating the correlation between certain antimicrobial exposure in patients and associated increase in abundance of bla_CTX-M_. The gene increase is then normalized to 16s abundance for overall internal QC for extraction efficiency and bacterial content. The work appears to be very well done and the methods and approach to analysis appear to support the conclusions of the authors. The figures are easy to interpret and additive to the manuscript (not sure if Figure 1 is critical but it is interesting to look at the variability across the data set and would favor keeping). Also the discussion includes a portion focused on relative biomass and bacterial presence in the gut of patients on antimicrobials but it is not clear they would be able to infer this information from the variability in swab collection and extraction efficiency. Would favor revising the discussion to address this issue.

One concern is the amount this manuscript overlaps with another previously published manuscript (Meletiadis, J. et al., 2017). The authors state in the Introduction section that they used a subset of the data but in reviewing the other manuscript it appears that there are 133 patients in this analysis and 122 from the other description. They also used very similar methodology by normalizing the bla_CTX-M_ abundance with 16s. I think it would be helpful for the authors to further explain how this differs from prior work and is additive to the literature. The conclusions from that manuscript were almost identical as well. Would favor a re-write to address this existing work and contrast the submitted manuscript.

Reviewer #3:

Niehus and colleagues present an original study in which they investigate the impact of different antibiotics on ESBL resistance within-host in *Enterobacteriaceae* carriers. The data comes from 133 patients from Romania, Serbia and Italy and consists in longitudinal series of rectal swabs (with a median of 5 swabs per patient). Swabs were analysed in order to quantify the abundance of bla_CTX-M_ and 16SrRNA. For both indicators, the within- and between-host variabilities, and the within-host dynamics are analysed using Bayesian state-space models. Parameters associated to the selection exerted by different classes of antibiotics on the dynamics of bla_CTX-M_, 16SrRNA and bla_CTX-M_/16SrRNA are estimated.

This is a very exciting and well written article. This is, to my knowledge, the first study attempting to analyse, using hypothesis-driven mechanistic models, the dynamics of within-host resistance genes and the impact of various classes of antibiotics. I believe that the study and results presented here represent a major contribution to the field.

– It would be good for the reader to detail and clarify how each indicator should be interpreted. My understanding is for example that a decrease in bla_CTX-M_/16SrRNA would more or less be interpreted as a decrease in resistance rate. However, it is not clear to me how important is the contribution of *Enterobacteriaceae* species to the global quantity of 16SrRNA.

– My main comment is related to the simulation study. It would be good to provide a validation of the model before running simulations, for example by doing out of fit assessment? Is that possible? Would you have enough power?

– It would be interesting to estimate whether the antibiotic classes have different delays of impact.

– I am not an expert but my understanding is that the aptitude to induce degradation of the flora may not be fully characterized by the "broad spectrum" characteristic. Other characteristics define whether the antibiotic destroys or not non-pathogenic (and anaerobic?) bacteria. It would be good to discuss more that aspect.

[Editors' note: further revisions were suggested prior to acceptance, as described below.]

Thank you for submitting your article "Quantifying antibiotic impact on within-patient dynamics of extended-spectrum β-lactamase resistance" for consideration by *eLife*. Your article has been reviewed by three peer reviewers, and the evaluation has been overseen by a Reviewing Editor and Wendy Garrett as the Senior Editor.

The reviewers have discussed the reviews with one another and the Reviewing Editor has drafted this decision to help you prepare a revised submission.

Summary:

Antimicrobial resistance is recognised by the World Health Organization and others as one of the most pressing threats to global health. However, our understanding of the ecological processes that link antimicrobial use to the emergence and spread of resistance is limited. In particular (and surprisingly for such a fundamental question), a quantitative understanding of the within-host processes relating clinical antibiotic exposure to within-patient resistance abundance is lacking. For bacteria that colonize the human gut, such within-host processes are likely to play a major role in mediating the relationship between antimicrobial use and the prevalence of resistance in the wider population. This paper focuses on resistance in *Enterobacteriaceae* conferred by extended-spectrum β-lactamase (ESBL) production. Such ESBL-producing organisms are responsible for a high and increasing burden of disease globally. The authors aimed to determine the effects of typical antimicrobial exposures in hospitalized patients on the dynamics of resistance gene abundance and total bacterial load. The authors asked: if such effects exist, how do they vary between different treatment regimens, and what is the predicted impact of antibiotic exposure on persistence of patient colonization with ESBL-producing organisms?

Essential revisions:

The manuscript remains of high interest because of the conceptual innovation of this work related to the model.

However, reviewers still do not fully understand the dynamical model, which in our view was one of the key points of novelty of this work because it makes the link between mechanistic processes and the noisy observations. Accordingly, further revisions of this section should be completed before the paper can be accepted for publication.

It would be helpful to link the data to the notation so we know which symbols correspond to data. For clarity, even after doing this please note which are the hidden-state variables (in the dynamic model; it's listed in the observation/process noise model).

Technically, the notation a, b refers to a closed interval so infinity should not be listed in this form. We would suggest simply the phrase "improper uniform prior".

Equation 4 and text: it appears that you haven't "added them on a log scale" because the sum that you have computed goes into the mean of the normal distributions in Equation 4.

What does x_t=1_ = x_t=1_a_0_c_1_c_2_ have to do with any of the terms in (4) ? The text in remains inscrutable compared to the equations and notation. There is mention of the model "looping" through time points – how does this work? Please expand the text and be explicit about the likelihood at each stage, what the looping steps are, whether the terms are added to the mean or are multiplicative, and so on. Please define x_i,j,g=ratio_, x_i,j,g=16S_ and all notation.

When you say that you "fitted the model with STAN" please give the posterior decomposition, and the likelihood. State which parameters are estimated. The notation of the model should be connected to Figure 4 and 5 where the model's inferences are shown.

Despite the response to review where you agree that it will be difficult to assess "abundance", you then include the following statement. It seems that the authors agree with the limitations around abundance in the response to review but then do not fully incorporate into the discussion. Please consider revising. Example: “Surprisingly, despite the relatively broad antibacterial spectrum of cefuroxime and ceftriaxone, there was no evidence that exposure to these antibiotics reduced 16S rRNA abundance (although we note that a "broad" spectrum is not defined in relation to the microbiota but to bacterial species of clinical importance).”

---

## [Author Response]

Essential revisions:The reviewers brought up the previous publication, Meletiadis, J. et al., 2017, in which a good portion of the cases were reported, and where some of the same conclusions were reached.The authors should clarify the novel contributions of this work in light of the previous manuscript. It seems that the key claimed results here are (1) variability, Figures 1, 2; (2) effect of treatment on relative resistance; (3) effect of treatment on abundance and (4) the model with fitting, and consequently the estimates re duration of carriage. Each has limitations.Please set these results, with the novelty and limitations, in context.

This is an important point. As we mention in our Introduction, a subset of the data we use were also used in a publication by Meletiadis et al., in 2017. Their work used standard statistical methods (Fisher exact test, classification regression tree, stepwise logistic regression) on collapsed data (each patient’s time series was converted to a zero or one) to ask, is there a statistical association between ceftriaxone treatment and bla_CTX-M_ normalised by total bacteria? Our study addresses a set of related but different questions and makes use of the full, uncollapsed data. First we are concerned with the differential effects of a range of antibiotics (not only one) both on bla_CTX-M_ and on total bacterial load. Second, our focus is on developing a mechanistic understanding of the impact of antibiotics on the dynamics of both bla_CTX-M_ and 16S rRNA, rather than considering static statistical associations. Third, through formulating and fitting a dynamic model to the data we aimed to estimate ecologically important parameters (strength of selection, speed of resistance waning) which are of interest in themselves. Finally, we sought to make predictions about the impact of different antibiotic treatment regimens on persistence of bla_CTX-M_. We have now changed the final paragraph of our Introduction to clarify how our goals are different from the previous work. We also added a new section in the Discussion that highlights the methodological novelties and advantages where we previously only stated the limitations.

The modelling is clearly new and I think it is important. In that direction, the clarity should be improved. I particularly noted the very long paragraph in subsection “Dynamic within-host model”, and the benefit that explaining the model and its assumptions (and limitations) for non-statisticians, non-modellers would likely bring.

We very much agree with this point. In response, we have rewritten the “Dynamic within-host model” section by breaking it into clearer sections, and by more clearly explaining the novelty of the model and more clearly listing the model assumptions. We also make sure to use language that is accessible for non-statisticians and by explaining some mathematical terminology.

Please clarify re interpretation of 16s, abundance as noted by reviewers.

As noted by reviewer 1, it would be interesting to validate 16S as an appropriate measure of cell density. Our study did not include an internal validation of the 16S rRNA measurements, however other studies have previously shown the ability of qPCR to quantify bacterial amounts in fecal samples (Rinttilä, 2004). We now mention this previous validation and the limitation that we do not include an internal validation in the “Quantitative PCR” section. We further added in our Discussion a comment on newer methods that quantify specifically viable bacterial cells (van Frankenhuyzen et al., 2013), which may guide future studies like ours.

Related to this, reviewer 2 notes that it may be difficult to draw conclusions about antibiotic effects on 16S rRNA given variability in DNA extraction efficiency. We agree that the variability of DNA extraction is expected to be large. Therefore our model was specifically designed to directly estimate variability that affects repeat measurements (through noise in the swab efficiency and DNA extraction) and to separate this variability from real effects on the underlying true 16S rRNA quantity. We realised that in our previous manuscript we only mentioned the swab procedure as a source of noise affecting the DNA concentration in the qPCR machine. We correct this now by being more explicit in various places for our new manuscript:

· “noise through the swab process and gene extraction”

· “observation noise (stochastic effects impacting the swab efficiency, DNA extraction or the qPCR measurements)”

· “variation in 16S rRNA would be caused mainly by the swab procedure and DNA extraction (and other steps in the protocol)”

· “error introduced by the swab procedure, DNA extraction etc.”

· “noise from qPCR measurement or from the DNA extraction process” “observation uncertainty – for instance through the swab procedure, DNA extraction, or qPCR process”

Reviewer 2 also states that one would anticipate antibiotics to cause shifts in relative species abundances as opposed to reductions in overall cell numbers. We fully share this expectation, in particular for antibiotics that target only a small subset of species present in the microbiome. Indeed, almost all of our inferred 16S effects support this hypothesis. We now take this up more explicitly in our Discussion by saying: The stable 16S abundance could be due to rapid overgrowth of non-susceptible strains which replace bacteria killed by these antibiotics (Hildebrand et al., 2019), leading to shifts in species composition rather than total species abundance.

Please improve Figures 1 and 2. While "hieroglyphics" may go a bit far, I agree that Figure 1 is visually striking but hard to see and interpret. Perhaps in Figure 1 you could group the time curves by similarity, or present an expanded view of some representative curves, with the full set in an appendix. Or something like that.

We have now changed Figures 1 and 2 to make them easier to interpret. In particular, we followed reviewer 1’s comment on Figure 2 and made it more concise by removing the additional bars of the lower part that indicated between and within patient time series variability. They were confusing as they showed a variability on the same scale as the sequence abundance, and between and within patient variability can easily be summarised and reported as numbers, which is what we do now.

For Figure 1, we have now produced a new version that contains a random selection of 50% of the study patients only. We additionally improved the previous figure with all patients by choosing a colourblind-friendly colour scheme and by increasing the plot area so that gene dynamics can be seen more clearly. We do agree with reviewer 2 (“this figure is interesting to look at the variability across the data set and would favor keeping”) that this version of the figure is a nice way to show the entire dataset. As reviewer 1 mentions, through eyeballing it appears impossible to spot waxing and waning in response to antibiotic treatment or to spot differences between treated and untreated patients, and this is exactly the purpose of this figure i.e. to show how noise is dominating the dynamics. Because we think that both are effective ways of presenting the data we attach both versions of Figure 1 for the editor to make the final decision on which one to include should the manuscript be accepted.

For ‘Figure 1 alternative’ the adjusted caption is: “Time series plots demonstrating the diverse range of dynamical patterns of bla_CTX-M_ resistance gene abundance across patients. For visualisation purpose, a subset of 49 patients was selected uniformly at random from all 132 patients with more than one rectal swab. The x-axis scale is identical across panels, the length of one week is given for scale in the top-left corner. Timelines are ordered by length. The y-axis scale differs between panels, with the space between vertical grey lines representing a 10-fold change in the absolute bla_CTX-M_ gene abundance (measured in copy numbers). The left-hand side shows patients who received antibiotic treatment (n=113), and the two right-hand side columns are patients without antibiotic treatment (n=19). For clarity, we show only the twelve most frequently used antibiotics in distinct colours and other antibiotics in light grey.”

Reviewer #1:The motivation for this study and its goals are certainly important and timely. The collateral effects of antibiotic treatment on the distribution of bacteria species and strains in the enteric flora (microbiome) of humans and the incidence of antibiotic resistance genes and resistance encoding plasmids are subjects of considerable importance, epidemiologically as well as clinically. In their Introduction, the authors do a fine job of presenting this and particularly so for the extended-spectrum β-lactamase (ESBL)-producing Enterobacteriacae and the blaCTX family of resistance genes, that are the focus of their study.This investigation certainly has the virtue of being extensive; fecal samples were taken sequentially from 133 hospitalized patients from three countries (Romania, Serbia and Italy) for a median of five samples from each patient. This virtue does have a downside; these patients were hospitalized for different reasons and treated orally of intravenously with 10 different antibiotics. The results presented suggest that some antibiotics are more likely than others to affect the distribution and abundance of these enteric bacteria and the bla_CRX-M_ genes, their study is correlative rather than mechanistic. From the results presented, it's not clear how the different modes and purpose of treatment the within-patient dynamics ESBL resistance. From what they say in their Discussion, the authors are very much aware of this limitation of this study.

Indeed, we are aware of the limitations of our observational data regarding causal conclusions, including a potential for selection bias and confounding through patient differences that affect both the treatment decision and resistance dynamics, as well as time-varying confounding. We now add a statement on this in the Discussion.

Although their observation some antibiotics are more likely than others to affect the enteric microbiome and the frequency of bla_CRX-M_ genes could have epidemiological and clinical implication, they don't really consider these potential implications of their results. It's also not clear how this information can be used.

Even though our Figure 5 shows the impacts of different alternative antibiotic choices on the duration of resistance carriage (a clinically interesting and relevant outcome) we refrain from drawing any clinical conclusions or recommendations from our results. This is because: 1) our study only looks at the effects on a single type of resistance; and 2) it is not clear what the individual patient or the community effects are of the observed changes to the bla_CTX-M_ dynamics.

Presumably the choice of antibiotics, mode and frequency of administration in these hospitalized patients is based on the nature of the infection and information about the susceptibility of the target bacteria to the drugs employed for treatment. While in an ideal world, consideration should to the collateral effects of treatment; we are far from that ideal world. This is particularly so for hospitalized patients. It would be of some interest to do an analogous study for the treatment of community-acquired infections (about 90% of antibiotic use in humans), where information about the collateral effects of treatment with different antibiotics may have broader and implications that could be implemented.

This point is well taken and the obvious reason for studying hospital patients is the relative ease of following up patients. There are however existing studies on the way that explore similar questions in a community setting.

Reviewer #2:This is an interesting and important piece of work demonstrating the correlation between certain antimicrobial exposure in patients and associated increase in abundance of bla_CTX-M_. The gene increase is then normalized to 16s abundance for overall internal QC for extraction efficiency and bacterial content. The work appears to be very well done and the methods and approach to analysis appear to support the conclusions of the authors. The figures are easy to interpret and additive to the manuscript (not sure if Figure 1 is critical but it is interesting to look at the variability across the data set and would favor keeping). Also the discussion includes a portion focused on relative biomass and bacterial presence in the gut of patients on antimicrobials but it is not clear they would be able to infer this information from the variability in swab collection and extraction efficiency. Would favor revising the discussion to address this issue.One concern is the amount this manuscript overlaps with another previously published manuscript (1). The authors state in the Introduction section that they used a subset of the data but in reviewing the other manuscript it appears that there are 133 patients in this analysis and 122 from the other description. They also used very similar methodology by normalizing the bla_CTX-M_ abundance with 16s. I think it would be helpful for the authors to further explain how this differs from prior work and is additive to the literature. The conclusions from that manuscript were almost identical as well. Would favor a re-write to address this existing work and contrast the submitted manuscript.

This is the first major point raised by the editor and we addressed this above.

Reviewer #3:Niehus and colleagues present an original study in which they investigate the impact of different antibiotics on ESBL resistance within-host in Enterobacteriaceae carriers. The data comes from 133 patients from Romania, Serbia and Italy and consists in longitudinal series of rectal swabs (with a median of 5 swabs per patient). Swabs were analysed in order to quantify the abundance of bla_CTX-M_ and 16SrRNA. For both indicators, the within- and between-host variabilities, and the within-host dynamics are analysed using Bayesian state-space models. Parameters associated to the selection exerted by different classes of antibiotics on the dynamics of bla_CTX-M_, 16SrRNA and bla_CTX-M_/16SrRNA are estimated.This is a very exciting and well written article. This is, to my knowledge, the first study attempting to analyse, using hypothesis-driven mechanistic models, the dynamics of within-host resistance genes and the impact of various classes of antibiotics. I believe that the study and results presented here represent a major contribution to the field.– It would be good for the reader to detail and clarify how each indicator should be interpreted. My understanding is for example that a decrease in bla_CTX-M_/16SrRNA would more or less be interpreted as a decrease in resistance rate. However, it is not clear to me how important is the contribution of Enterobacteriaceae species to the global quantity of 16SrRNA.

This is an interesting and important point, and our previous version of the manuscript was not clear enough about why we chose to model effects on bla_CTX-M_ / 16S rRNA. As the reviewer suggests, we expect the bla_CTX-M_ / 16S rRNA ratio to be approximately propotional to the fraction of bacterial cells harbouring the bla_CTX-M_ gene. We then model antibiotic effects on the ratio bla_CTX-M_ / 16S rRNA (representing effects on the relative fitness of bla_CTX-M_ carriers) and effects on the total bacterial abundance as approximated by 16S. But our model also gives the resulting dynamics of absolute bla_CTX-M_ abundance, which is important for predicting its extinction and persistence.

– My main comment is related to the simulation study. It would be good to provide a validation of the model before running simulations, for example by doing out of fit assessment? Is that possible? Would you have enough power?

We fully agree that model validation is important. In our study we therefore performed model comparison in which we compared our model with a model without antibiotic effects – we find that the data supports effects of antibiotics. For this we now performed a leave-one-out cross-validation approximation. We approximate because the number of patients treated with the same antibiotics is small. We have revised the manuscript so that we explain this more clearly now.

– It would be interesting to estimate whether the antibiotic classes have different delays of impact.

This is an interesting point that is on our (long) list of possible extensions to our model. However, while interesting, we consider this to be outside the focus of this study and leave this question to further work on this and similar datasets.

– I am not an expert but my understanding is that the aptitude to induce degradation of the flora may not be fully characterized by the "broad spectrum" characteristic. Other characteristics define whether the antibiotic destroys or not non-pathogenic (and anaerobic?) bacteria. It would be good to discuss more that aspect.

This a great point: the term “broad”-spectrum describes an antibiotic’s killing-spectrum in terms of known clinical pathogens. This may or may not be related to the extent of impact on the entire microbiota biomass. We now clarify this in the Discussion (first paragraph).

[Editors' note: further revisions were suggested prior to acceptance, as described below.]

Essential revisions:The manuscript remains of high interest because of the conceptual innovation of this work related to the model.However, reviewers still do not fully understand the dynamical model, which in our view was one of the key points of novelty of this work because it makes the link between mechanistic processes and the noisy observations. Accordingly, further revisions of this section should be completed before the paper can be accepted for publication.It would be helpful to link the data to the notation so we know which symbols correspond to data. For clarity, even after doing this please note which are the hidden-state variables (in the dynamic model; it's listed in the observation/process noise model).

We have rewritten major parts of the dynamic model description. First, we have explained all symbols in Equation 4 within the paragraph below (previously symbols where explained in two separate parts). Second, we have included a new paragraph that lists all the symbols that represent data, and all the symbols representing estimated variables. We have been explicit about which symbols denote hidden-state variables.

Technically, the notation a, b refers to a closed interval so infinity should not be listed in this form. We would suggest simply the phrase "improper uniform prior".

This is a good point, and we now use the phrase “improper uniform prior” instead.

Equation 4 and text: it appears that you haven't "added them on a log scale" because the sum that you have computed goes into the mean of the normal distributions in Equation 4.What does x_t=1_ = x_t=1_a_0_c_1_c_2_ have to do with any of the terms in (4) ? The text in remains inscrutable compared to the equations and notation. There is mention of the model "looping" through time points – how does this work? Please expand the text and be explicit about the likelihood at each stage, what the looping steps are, whether the terms are added to the mean or are multiplicative, and so on. Please define x_i,j,g=ratio_, x_i,j,g=16S_ and all notation.

We agree that this part was not well explained. In response, we have substantially changed this part of our model description. The new description is explicit about which parameters are on a log-scale, and what the real scale reflects (gene copy numbers). To clarify the way antibiotic effects work, we give a numerical example, in which we used the same symbols as our Equation 4, instead of introducing new ones. We have also clarified how the sums of the definition of *f(abx*) work, and we explain each likelihood term in our model, and also give the full posterior distribution (see below). Finally, we have defined all notation, including x_i,j,g=ratio_, x_i,j,g=16S_.

When you say that you "fitted the model with STAN" please give the posterior decomposition, and the likelihood. State which parameters are estimated. The notation of the model should be connected to Figure 4 and 5 where the model's inferences are shown.

It was not entirely clear to us what was meant by ‘posterior decomposition’. Based on what Neiswanger et al. (https://arxiv.org/pdf/1510.04163.pdf) refer to as posterior decomposition, we now (in addition to Equation 4) give the full posterior distribution as the product of different terms (Equation 5). But in addition we also explain our three different likelihoods in terms of decomposing the total amount of variation in the data into variation attributable to different processes. We finally also added the marginal posterior distributions (together with prior distributions) of the antibiotic effect parameters (c_z,b_) as an additional Supplementary Figure. To address the second point, we now explicitly list all parameters of Equation 4 that are estimated. Finally, to link methods to figures, we added an explanation of how the posterior draws of c_z,b_ were transformed before plotting them in Figure 4. We added a reference to Figure 4 at that point, and we added two references to Figure 5 into the description of the posterior predictions.

Despite the response to review where you agree that it will be difficult to assess "abundance", you then include the following statement. It seems that the authors agree with the limitations around abundance in the response to review but then do not fully incorporate into the discussion. Please consider revising. Example: “Surprisingly, despite the relatively broad antibacterial spectrum of cefuroxime and ceftriaxone, there was no evidence that exposure to these antibiotics reduced 16S rRNA abundance (although we note that a "broad" spectrum is not defined in relation to the microbiota but to bacterial species of clinical importance).”

Based on this comment, we have now decided to remove any interpretation of antibiotic effects on 16S gene abundance. We think that this also helps the discussion be more focused on the CTX-M resistance gene.